# Clinician's perspectives on gene therapy for Alzheimer's disease: A qualitative study

**Lilly Kelemen**[1], **Ishika Gupta**[1], **Zollie Yavarow**[1], **Samantha I. Smith**[2], **Kim G. Johnson**[3], **Nathan A. Boucher**[2,4] *

1 Duke University, Durham, NC, United States of America, 2 Sanford School of Public Policy, Duke University, Durham, NC, United States of America, 3 Department of Neurology, School of Medicine, Duke University, Durham, NC, United States of America, 4 Division of Geriatric Medicine, School of Medicine, Duke University, Durham, NC, United States of America

\* Nathan.boucher@duke.edu

## Abstract

### Introduction

We aimed to understand clinician views regarding gene therapy as a future treatment for Alzheimer's disease (AD) and potential barriers and facilitators to its use.

### Methods

We interviewed ten clinicians who treat patients with AD. Clinicians helped design a semi-structured interview including the following domains: establishing understanding, cost/access, quality of life, and religion/spirituality. Transcripts were analyzed by a coding team using descriptive content analysis with inductive approach.

### Results

Clinicians identified three main areas of concern: 1) potential clinician and patient understanding of gene therapy and Alzheimer's disease 2) consideration of inequity (i.e., care access, disease awareness along with education level, family support, trust in care systems); and 3) considerations in decision-making (i.e., religious/spiritual beliefs and method of treatment delivery as a decision-making tools).

### Discussion and conclusion

Findings highlight areas for knowledge-building for patients and clinicians alike. Clinicians must be aware of patient/family educational needs and gaps in their own clinical knowledge before engaging patients/families with new technology. Allowing time for questions is crucial to building rapport and trust.

## Introduction

Alzheimer's disease (AD) is a progressive neurodegenerative disease causing cognitive decline and functional loss which presents challenges foremost for patients and their families, but also

**Data Availability Statement:** Regarding data repository filing, this is a qualitative data set and would be of limited reproducible value especially since it would have to be fully anonymized.

Additionally, participants did not give consent to data sharing beyond the study team. Contact for IRB questions: campusirb@duke.edu. This is an expert led student project. We appreciate your consideration.

**Funding:** The author(s) received no specific funding for this work.

**Competing interests:** The authors have declared that no competing interests exist.

for society and for the health care system [1]. An estimated 6.2 million people are living with Alzheimer's disease in the United States. That number is expected to increase to a staggering 14 million by 2060 [2].

While reported deaths from heart disease, stroke, and HIV decreased from 2000 to 2019, deaths caused by AD increased more than 145%. The COVID-19 pandemic may have contributed to an even larger increase in AD deaths from 2019 to 2022 [2]. AD pathology is diagnosed by the A,T,N classification system developed and revised by the NIA-AA [3]. To fulfill criteria for pathologic pre-clinical AD, there must be the presence of β-amyloid plaques (A), +/- neurofibrillary tau deposits (T) and +/- neurodegeneration (N). Because amyloid plaques build up slowly, it can take years before symptoms become recognizable leading to a clinical diagnosis of Alzheimer's disease.

There are few effective treatment options for neurodegenerative diseases such as AD. Traditional treatments for AD focus on treating symptoms as opposed to reversing disease progression [4]. Drugs with disease-modifying potential that reduce amyloid-beta plaques in the brain of patients with Alzheimer's disease have recently received FDA approval (Aducanumab accelerated approval in June 2021; Lecanemab accelerated approval in January 2023 and traditional approval in July 2023) [5, 6]. However, their effectiveness is also limited and they are not a cure. Removal of amyloid from the brain by lecanemab slows progression of cognitive decline by 27% over 18 months but does not halt progression of cognitive decline.

Therefore, as rates of AD and AD-related deaths increase in the population with devastating consequences, there is high demand from patients, caregivers, and patient advocates for effective disease-modifying therapies. Gene therapy is an emerging disease-modifying therapy with interesting potential in many disease states. In November 2023, the U.S. Food and Drug Administration approved the first cell-based gene therapies, Casgevy and Lyfgenia, to treat sickle cell disease [7]. These approvals mark an exciting advancement to expand the possible treatment options for patients living with life-altering diseases. Researchers are working to develop similar gene therapy treatments to alter the underlying genetic mechanisms of Alzheimer's disease and decrease genetic risk rather than treat symptoms as they appear.

There are several known genetic mechanisms involved with AD that present potential targets for gene therapy including the Amyloid Precursor Protein (APP), Presenilin 1 (PS1) and Presenilin 2 (PS2) mutations which result in autosomal dominant inheritance of early onset AD and the apolipoprotein E4 (APOE4) allele, a genetic risk factor which increases genetic risk for late onset AD, also known as LOAD [8]. The ε4 isoform of the apolipoprotein E (APOE) gene imparts increased genetic risk for AD and has been linked with varying degrees of amyloid build up in the brain [9]. Every individual carries two alleles of the APOE gene, either an of three possibε2, ε3, and ε4, with the most common allele being ε3. Therefore, there are six possible APOE genotypes including ε2/ε2, ε2/ε3, ε3/ε3, ε2/ε4 ε3/ε4 ε4/ε4. While the rare ε2 allele may actually decrease risk of developing AD later in life, carrying the ε4 allele is the strongest genetic risk factor in developing LOAD [10]. Carrying one copy of APOE-4 increases risk of developing AD about twofold, yet having two copies increases the risk up to twelvefold. Therefore, APOE4 represents a potential therapeutic target for gene therapy in Alzheimer's disease that could affect a large number of people in the population [11]. In fact there is currently a trial using APOE as a gene therapy target [12].

The potential for disparities of care in the process of how clinicians educate AD patients and their families about complex new treatments and engage in joint clinical decision-making cannot be understated. Lower education levels, higher rates of poverty, and greater exposure to adversity and discrimination are all risk factors that increase the likelihood of dementia [2] and create disparities in dementia diagnosis and treatment that are well documented. American racial minority groups are often diagnosed at a later stage of disease than White Americans [13] and it is well known that Black Americans are less likely to be prescribed or use a cognitive

medication for dementia than White Americans. Disparities in AD rates are further complicated by a significant underrepresentation of Black Americans in AD research studies, partly explained by general distrust of the American healthcare system due to its history of abuse of racial minority patients [14, 15].

Patient and caregiver perceptions of treatment options are important. There is previous research detailing caregivers and patients' perception of nonpharmacological interventions in adults with AD; however, research exploring how patients and caregivers understand gene therapy as an emerging treatment for AD is lacking [16]. Therefore, the role of the clinician may be an important factor in patient and caregiver understanding of gene therapy.

This study explores clinicians' views on gene therapy for AD, specifically targeting the APOE4 allele, a treatment modality currently in clinical trials. In this study, wee took a high-level approach, seeking to understand how clinician's themselves perceive this particular AD gene therapy and their understanding of the possibilities of gene therapy, both factors which affect their ability to educate patients. We hypothesize that more thorough knowledge of barriers and facilitators to patient and clinician understanding of gene therapy could lead to better informed medical decision-making that is shared between both clinician and patient [17].

## Methods

This exploratory study is formative work to optimize how emerging gene therapy technologies are introduced to both medical care providers as well as healthcare consumers. We conducted semi-structured interviews with academic center clinicians who would guide patients through decisions regarding gene therapy for Alzheimer's disease. The [redacted] IRB reviewed and approved this research study and the consolidated criteria for reporting qualitative research (COREQ) was completed.

### Participants

To recruit clinicians, we collaborated with department heads who suggested clinicians appropriate for the study across disciplines, who work with patients in the setting of Alzheimer's disease treatment. Inclusion criteria included experience as a healthcare provider advising and treating those living with dementia. Participants completed a phone screening to confirm they met study inclusion criteria and were then scheduled for a virtual video interview following an explanation of the study and verbal consent to participate. No monetary compensation was provided. All approached clinicians consented and completed the interview following provision of study information and the opportunity to ask questions.

### Measurements

The multidisciplinary research team created a semi-structured interview guide (see Supplement) complete with open-ended interview questions and probes. Three clinicians with social work and neurology expertise, knowledge of Alzheimer's disease, and knowledge of research methods in our health system reviewed the guide and helped the team identify domains of interest to our health system. NAB piloted the interview guide with two clinicians to improve understanding and word clarity. We incorporated feedback from both engagements.

Participants were first asked about their current level of understanding of gene therapy. Interviewers then followed this initial question by defining gene therapy for each participant with the same standard definition: "Gene therapy is an experimental technique that uses genes —a unit of heredity written in our DNA—to treat or prevent disease. In the future, this technique may allow health care providers to treat a disorder, such as Alzheimer's disease by adding or changing a gene in a patient's cells."

Along with demographic questions, domains queried in the interview guide included: cost, access, and quality [18]–predominant tensions driving healthcare delivery––as well as potential patient and clinician concerns around establishing understanding of gene therapy as a treatment. Additionally, we asked about the role of religion and spirituality, as our interview guide reviewers indicated these as important considerations in patients' health decision making locally. Audio recorded interviews were between one trained interviewer (either LK or IG) and one enrolled clinician, without observers. Interviewers were undergraduate students and trained by NAB, an experienced interviewer. The resulting auto transcription was checked for accuracy by the interviewer immediately following the interview.

## Analysis

When further observations and analysis were no longer generating new or discrete themes, the study team stopped interviews and recruitment [19, 20]. Two coders conducted a descriptive content analysis with an inductive approach using larger team meetings to help develop the codebook [21]. Coders IG, LK, and SS used NVivo 12 Pro (QSR International Pty Ltd. 2018) to manage data and develop codes to segment participant interview data into conceptual categories (e.g., incorporating text describing similar concepts).

We identified content-driven codes and applied them to the text for each of the conceptual categories (e.g., potential themes related to patient or provider education). Coders selected three interview transcripts, and two coders separately identified themes. After reconciling emergent themes and revising the codebook as a larger team, two coders separately coded the next three interviews. The team discussed the application of coding for those three interviews. Confirmation coding was used thereafter (coded by one, confirmed or disputed by the second). Coding disagreements were resolved through meetings with the larger research team. SS and IG then completed coding on all transcripts. We examined code frequencies across transcripts to identify salient factors for participants. To conclude the analysis, we used direct quotes (when able) and indirect quotes to demonstrate common themes.

## Results

We conducted n = 10 (no refusals or drop-outs) in-depth interviews ranging from 20 minutes to approximately 60 minutes with academic center clinicians working with patients living with Alzheimer's disease. **Table 1** shows the characteristics of this clinician sample. Analysis resulted in a focus on three main themes with related subthemes– **1)** *Understanding of gene therapy and Alzheimer's disease* **2)** *Consideration of inequity* (care access, disease awareness along with education level, family support, trust in care systems); and **3)** *Considerations in decision-making* (religious/spiritual beliefs and method of treatment delivery as a decision-making tools). Illustrative quotes for each subtheme are included.

### Potential gaps in understanding of gene therapy and Alzheimer's disease

Our clinician interviewees noted that, to provide optimal care to people living with Alzheimer's disease (AD), healthcare providers should consider patient characteristics. This applies to the patient's preferences for their own quality of life and the patient's current level of participation in activities meaningful to them. For example, what is their level of functioning and communication––can they perform activities of daily living, or are they confined to a bed? A provider's keen awareness of the degree of AD and any concomitant illness (and current medications) also allows the provider to have a global view of their patient's health and where new treatments may fit. Expectedly, providers expressed a need to know the risks and benefits and the cost of a new treatment before recommending it to a patient.

**Table 1. Sample characteristics.**

|  | (*N* = 10) |
|---|---|
| **Gender, *N* (%)** |  |
| Male | 3 (30) |
| Female | 7 (70) |
| **Age (years), *N* (%)** |  |
| 30–40 | 3 (30) |
| 41–50 | 1 (10) |
| 51–60 | 5 (50) |
| 61+ | 1 (10) |
| *M* (*SD*); range | 49.8 (11.14); 33–69 |
| **Race** |  |
| White | 8 (80) |
| Latinx | 1 (10) |
| Asian/Pacific Islander | 1 (10) |
| **Highest education level, *N* (%)** |  |
| Master's Degree | 4 (40) |
| Doctorate Degree | 6 (60) |
| **Clinical discipline, *N* (%)** |  |
| Physician | 4 (40) |
| Physician Assistant | 1 (10) |
| Nurse Practitioner | 2 (20) |
| Principal Investigator/Research/Professor | 3 (30) |
| **Years working with dementia populations** |  |
| <5 | 1 (10) |
| 5–10 | 4 (40) |
| 20–30 | 4 (40) |
| 30+ | 1 (10) |
| *M (SD);* range | 14.85 (11.6); 1.5–37 |
| **Care setting (more than one setting for n = 3)** |  |
| Inpatient | 3 |
| Outpatient | 8 |
| Telehealth | 2 |

*Notes*: M = mean SD = standard deviation

❖ *". . . as they continue to progress, and they've lost all of their ADLs [activities of daily living], at that point it's progressed to such a degree that it seems that any further treatment would just be burdensome and so I wouldn't initiate anything at that point."*

❖ *"If it alleviates some symptoms but doesn't really stop the disease that's a whole different question as opposed to something that could actually reverse the disease. . ."*

Interviewees also discussed helpful education strategies to target healthcare providers treating people living with AD. These educational resources included research conferences reviewing best practice and new developments in the field, consulting with knowledgeable colleagues with more experience, and looking to institutions within which they practice for guidance on currently approved AD management.

Access to information is important for patients, according to the interviewees, so patients and families can make informed decisions. The education strategies that providers and health

systems use to engage patients should be comprehensible and delivered with accessibility in mind. These may take the form of digestible introductions to the topic and resources that can be reviewed over time. When and where those conversations take place––and what type of provider delivers the information––are concerns.

❖ *". . . TV and newspapers and magazines tend to be very superficial. . . 'this is the next cure for this particular disease without recognizing all of the limitations."*

❖ *". . . there could be, you know, education pamphlets and things like that in waiting rooms, but I think it would be a lot to expect within the time frame of a primary care visit to be able to start to introduce these topics. I do talk often with patients and families about genetic risk factors in the course of my appointment, takes quite a bit of time for people to understand it. . ."*

❖ *". . . providing education to [patients and caregivers] would be helpful. Sometimes there are some caregivers that like a lot of information. . . give them the recent research or study that was conducted that they can access themselves online potentially. . . probably some sort of handout or fact sheet. . . to the actual studies."*

Patients and families, according to interviewees, should also be informed of the effects of treatment, such as the intended and unintended side effects and the goal of treatment. Patients should clearly understand at what stage the treatment can be applied and when it might not be beneficial. Their healthcare provider best informs this clarity and should involve close family members in the discussion.

❖ *". . . in what ways would [the treatment] affect other parts of the brain or body, besides the area that it's targeting or what it's meant to do."*

❖ *"I'd work with the caregivers. . . we have to look at patients and caregivers as having autonomy, decision-making capacity. . . and work with them to explain the risk versus the benefits."*

Patients should also have knowledge of how a drug or other intervention is administered as it may have implications for pain or discomfort, time needed for clinic visits, or other effects on lifestyle. Interviewees note patients will have many questions; they will want to know details.

❖ *"Is it cumbersome? Is there going to be a lot of follow up bloodwork or imaging? Um, is it painful? What's the time commitment––is it a one-time infusion or is it a recurring treatment? Is it going to reverse any disease that's already set in or is it going to prevent to further deterioration? All of that."*

## Consideration of inequity

Interviewees were concerned about access to care, specifically access to a high technology intervention like gene therapy, which may require many resources limited to tertiary care centers. Cost, as a barrier to access, was also highlighted as a likely constraint for many patients, presuming that advanced care such as gene therapy would require substantial financial resources.

❖ *"In general, people with poor access to healthcare will wait longer before being seen for anything because it's not, you know, worth the money and the time away from other activities that actually generate money."*

Our clinicians cautioned that patients' awareness of their disease progression should be assessed in tandem with their level of education to ensure a fair and thoughtful provider-to-patient approach. Health literacy and numeracy would be perennial challenges especially in the context of something as complex as gene therapy.

❖ *"It's difficult because you've got such a wide range of levels of knowledge and so many people barely know what DNA is and don't really have a concept of how you would manipulate DNA in order to accomplish gene therapy."*

Family support would be a key consideration when working with patients as most of the care coordination and decision-making happens with family involvement.

❖ *"Caregivers play a huge role, oftentimes [if] the patient is cognitively impaired or in the mild dementia category they really just defer to the caregiver to make those decisions, so I think, you know, caregivers. . . kind of make a decision about whether or not we're going to do. . . advanced diagnostic testing, whether or not we're going to try this medication."*

Trust in healthcare providers and healthcare systems is also a consideration. Interviewees recalled that patients have varying levels of trust, and that people of color may be particularly mistrustful of unfamiliar treatments given a terrible legacy of mistreatment of people of color by the American medical system.

❖ *". . .there's a lot of distrust with the healthcare system or providers as well, that would prevent [Black patients] from seeking out care."*

❖ *"Highly educated white or Black [people] are more likely to participate in research studies, I think people who have less education are less likely to participate. I think there is a concern that maybe we're trying to pull something over on them, certainly, you know, Black patients, given the history of. . . research trials that were exploited."*

## Considerations for decision-making

Interviewees discussed common patient considerations for decision-making in treatments for AD. This included the role of spirituality and religion in patients' lives and its influence on medical decision-making. Additionally, the method of treatment delivery appeared to be another important consideration for decision-making.

Some patients, the interviewees noted, use spirituality or religion as a decision-making tool in the context of illness. While taking into consideration advice from a healthcare provider, they might also temper that understanding or their next step decisions with their understanding of a perceived higher power or religious guidance.

❖ *". . . common is the: 'You know, I prayed to God and he told me to do this' or that 'Miracles happen, and you know, this is going to help.' To help the study drug work better, they oftentimes appeal [to] or report that that God is working in their in their court."*

❖ *"I would say [religion/spirituality] is not necessarily something that's discussed out right, but maybe more so when you particularly asked about it, then they'll open up, and you know, they have no problem sharing it. . . I'd say more often than not, it doesn't come up unless you specifically asked."*

❖ *"I think a lot of it has to do with. . . how much control they have over their life and so if they kind of feel like well whatever is gonna happen is gonna happen or God has created the path. . . I think it depends on like how much perceived kind of control they have over their health, and how much they let their spiritual beliefs or their religion play a role in that."*

❖ *"I would say a pretty big role. . . their religious practices or spirituality sometimes can influence whether they pursue certain treatment recommendations or not."*

The interviewees identified a treatment's method of delivery as another potential consideration for patients. A treatment requiring multiple visits to a clinic may deter a patient who does not have reliable transportation, for example. The ease and pain level of gene therapy delivery (i.e., pill, infusion, injection) may be an important decision-making factor for the patient and their family caregivers.

❖ *"If they're not able to drive and if this requires something like weekly appointments, they're going to forego treatment because they just don't have the ability to go to all the appointments."*

❖ *"Is it a pill? Is it an infusion? The burden of administration on the patient would be a consideration. . . what is their access to getting the treatment easily.*

❖ *"I would hope, [the treatment is] not a painful option, not too tiresome. . . I think it has to be a limited burden [so] that it doesn't tire them out too much [or] hurt too much, [or cause] too much frustration."*

## Discussion

We explored the perspectives of clinicians regarding the future of AD gene therapy specifically targeting the APOE4 allele. We chose to focus on gene therapy targeting the APOE4 allele since this therapy is in the clinical trial stage; however, we recognize that attitudes regarding gene therapies to other potential targets including the APP, PS1 and PS2 genes would be a valuable addition to the analysis of attitudes toward gene therapy in AD.

While colleagues work diligently on the biological science of gene therapy, this study posited that many barriers potentially exist to the successful rollout of gene therapy in the practical world of patient care. Initial gene therapy trials will need to assess safety and efficacy in patients who have Alzheimer's disease, a population of patients with unique symptoms and potentially limited capacity for fully understanding the treatment, which could present barriers. Clinician education is important to the process to fill gaps in their clinical knowledge and therefore, enable them to anticipate patient's needs. Awareness of barriers upfront may allow early attunement to patient needs. Tailored care approaches may ensure person-centered treatment in a complex Alzheimer's disease population where neuropsychiatric symptoms and quality of life are major considerations [17, 22].

Our clinician interviewees spoke about patient/family-level considerations for applying new treatments like gene therapy. Accessible information sources like TV and newspapers rarely recognize the limitations of emerging treatments. It has been found that another patient group receiving advanced therapeutics (hematopoietic stem cell transplantation for sickle cell disease) primarily learned about treatments though social media, social networks, the news, and media—precisely the sources our clinicians cited as "superficial," with patients often not recognizing the limitations of these resources, in their opinion [23]. In an effort to educate patients in an accessible way, it may be helpful for clinicians to recognize the education potential of these media outlets and levy them to build patient knowledge.

Clinicians in our study clearly identified the importance of assessing patient decision-making capacity by better understanding the needs, disease state, and knowledge levels of both the patient and their caregiver. A 2012 study of dementia caregivers identified understanding the individual's needs, exploring potential options, and assessing their ability to make a choice as key components of the decision-making process [24]. This previous caregiver study confirms that the decision-making considerations highlighted by caregivers largely align with the responses of the healthcare providers in our study.

Inequity was a dominant topic in our study as our interviewees noted potential challenges in patients' healthcare access, education level, awareness of their disease, existing level of family support, and the perennial issue of trust in healthcare systems. These topics are present in the current literature, including the importance of family involvement, clinician-patient communication, and access to specialty care in dementia contexts [25]. Recent studies in the context of COVID-19––an amplifier of gaps in complex healthcare populations––point to patients' desire for more evidence-based information to make decisions and the importance of access to different treatment options [26]. Regarding inequities and trust among minority groups, Sloan et. al––studying coordination of dementia care in Black patient populations––noted Black participants expressed difficulty accessing adequate dementia information and care supports [27]. The same study noted a lack of effective communication about dementia symptoms and treatment in Black communities. The clinicians interviewed in this present study built on this idea, explaining that common mistrust of healthcare systems might also be a barrier to medical decision-making.

Consideration of religious and spiritual needs is paramount in our diverse patient populations; many consider these types of beliefs as necessary for enduring a disease process or deriving meaning from a challenging illness [28–31]. Religious faith or spirituality may be beneficial to facing the challenges of serious illness as patients may draw upon peace, hope, support, and even decision-making guidance from related beliefs [31, 32]. Our interviewees' opinions aligned with much of the literature on this topic: patients are known to consider a higher power and religious belief in how they approach their healthcare decisions [30].

Limitations include a focus on a specific health system in one area of one state and seeking views of only clinicians as opposed to including opinions of patients or caregivers. We recognize that the sample size of 10 clinicians, while adequate for qualitative research, gives a limited perspective on attitudes towards gene therapy and cannot be generalized to include all clinicians. Geography, payer mix, cultural differences, and political climate will play a role in uptake of any future gene therapy. The purpose of the study was to gather qualitative insights to further exploration of this topic.

It would have been ideal to also explore perspectives of people living with dementia or their family caregivers to impart more sophisticated dimensions on the topic––a possible area for future inquiry. In addition, the study methods assumed only patients with an Alzheimer's disease diagnosis would receive gene therapy treatment. We recognize that if gene therapy shows efficacy, the goals of future treatment may be to treat people with a copy of the APOE4 allele before symptoms develop. However, we feel it is important for clinicians and patients with Alzheimer's disease to consider gene therapy as a treatment since any clinical trials will start in a patient population affected with Alzheimer's disease due to ethical reasons. The first goal of gene therapy studies is to prove that the treatment is safe and it would be unethical to expect people with normal cognition to undertake the risk of a clinical trial to assess safety. If gene therapy is proven safe and clinical trials are expanded, further studies of clinicians' perspectives in a population of younger and normal cognition patients would be of interest.

## Conclusion

Our study adds early and necessary considerations for the application of gene therapy to AD. Indeed, this information is applicable to other AD therapies and other medical applications of gene therapy that may be initially tested for safety in an AD population. Our study shows that clinicians working in this area identify challenges ahead and readily suggest what might be done to mitigate them. Considering equity, family involvement, allowing time for patient/family education, and focusing on trust building and rapport are all facilitators to overcome the identified barriers. Gene therapy, with potential therapeutic expense and burden unlike many other clinical interventions, may require additional focus from clinicians, patients, and families.

When the technology becomes readily available, healthcare practices must find ways to reduce the perceived complexity of gene therapy and amplify its benefits in an understandable way for clinicians, persons living with Alzheimer's disease and their families. Cogent approaches to rolling out this new type of therapy will need to consider what clinicians and patients alike need to know, while also considering equity in providing care to diverse patient populations.

## Supporting information

**S1 File. Institutional review board approval.**
(PDF)

**S2 File. Provider interview guide.**
(DOCX)

## Author Contributions

**Conceptualization:** Lilly Kelemen, Ishika Gupta, Zollie Yavarow, Samantha I. Smith, Kim G. Johnson, Nathan A. Boucher.

**Data curation:** Lilly Kelemen, Ishika Gupta.

**Formal analysis:** Lilly Kelemen, Ishika Gupta, Zollie Yavarow, Samantha I. Smith, Nathan A. Boucher.

**Funding acquisition:** Ishika Gupta.

**Investigation:** Lilly Kelemen, Zollie Yavarow, Nathan A. Boucher.

**Methodology:** Lilly Kelemen, Zollie Yavarow, Kim G. Johnson, Nathan A. Boucher.

**Resources:** Nathan A. Boucher.

**Supervision:** Kim G. Johnson, Nathan A. Boucher.

**Writing – original draft:** Lilly Kelemen, Zollie Yavarow, Nathan A. Boucher.

**Writing – review & editing:** Lilly Kelemen, Ishika Gupta, Zollie Yavarow, Samantha I. Smith, Kim G. Johnson, Nathan A. Boucher.

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
