## [Decision Letter · Decision Letter 0]

24 Apr 2024

PONE-D-24-09705Gene Therapy for Alzheimer’s Disease: Clinicians’ perspectives on patient needsPLOS ONE

Dear Dr. Boucher,

Thank you for submitting your manuscript to PLOS ONE. After careful consideration, we feel that it has merit but does not fully meet PLOS ONE’s publication criteria as it currently stands. Therefore, we invite you to submit a revised version of the manuscript that addresses the points raised during the review process. After careful consideration by 2 Reviewers and an Academic Editor, all of the critiques of both Reviewers must be addressed in detail in a revision to determine publication status. If you are prepared to undertake the work required, I would be pleased to reconsider my decision, but revision of the original submission without directly addressing the critiques of the Reviewers does not guarantee acceptance for publication in PLOS ONE. If the authors do not feel that the queries can be addressed, please consider submitting to another publication medium. A revised submission will be sent out for re-review. The authors are urged to have the manuscript given a hard copyedit for syntax and grammar.

We look forward to receiving your revised manuscript.

Kind regards,

Stephen D. Ginsberg

Section Editor

PLOS ONE

Journal Requirements:

[No].

Reviewers' comments:

Reviewer's Responses to Questions

**Comments to the Author**

1. Is the manuscript technically sound, and do the data support the conclusions?

Reviewer #1: Yes

Reviewer #2: Partly

2. Has the statistical analysis been performed appropriately and rigorously? 

Reviewer #1: N/A

Reviewer #2: N/A

3. Have the authors made all data underlying the findings in their manuscript fully available?

Reviewer #1: Yes

Reviewer #2: No

4. Is the manuscript presented in an intelligible fashion and written in standard English?

Reviewer #1: Yes

Reviewer #2: Yes

5. Review Comments to the Author

Reviewer #1: Much of your work may be applicable to the use of monoclonal antibodies. Would you consider doing a similar study for that indication?

More importantly, your work on gene therapy for AD is more applicable in the near term to APP, PS1 and PS2 mutations. This could be mentioned in your discussion. Consider working with clinicians participating in the DIAN cohort studies and persons at risk of carrying such genes. You may get different answers than from your current study.

Reviewer #2: Overall

The researchers aimed to understand clinicians’ perspectives regarding gene therapy, their understanding of patients’ views, and potential barriers and facilitators to its use as a future treatment for Alzheimer’s disease (AD).

Given advances in developing these therapies and the lack of knowledge about clinicians’ and patients’ perspective, this is a topical study. It’s important to include the perspectives of the people who might be affected by new innovative treatments at an early stage.

The researchers identified gaps in knowledge, a need for educational materials, and barriers such as cost, difficulty to understand such complex interventions and lack of trust in health care systems.

However, my main concerns are:

- It remains unspecified what is meant by “gene therapies”, for whom this are intended, in what stage, etcetera. Although I understand that this may vary somewhat, by leaving it so vague, the findings could be applicable to any new medical intervention for Alzheimer’s disease.

- The sample population is small (10 health care providers) and not very representative (4 have a Master’s Degree as highest education). Based on this I doubt to what extent these findings may reflect and be generalizable to the perspectives of the clinicians who would discuss gene therapies with patients if and when they are introduced.

- In addition, potential patients are referred to as individuals who are cognitively impaired, and potentially unable to perform activities of daily living, to the extent they are no longer competent to make their own decisions. To the best of my knowledge, these therapies are generally intended for patients in earlier stages of the disease, where these challenges play a smaller role and other barriers apply (for example, related to treating relatively healthy individuals with invasive and risky therapies).

- In relation to this, the interview guide does not seem to be based on literature or findings, and only incorporates feedback from three clinicians with expertise in social work and neurology. The questions focus on specific topics, such as understanding, costs and tensions in health care delivery which correspond directly to the findings. So, I don’t feel like this is an exploration of which concerns, barriers and facilitators might be involved, but a specification of certain pre-specified aspects.

- Lastly, as this qualitative study is not reported in accordance to the Consolidated criteria for reporting qualitative research (COREQ) checklist or Standards for reporting qualitative research (SRQR) checklist (the authors don’t report adhering to guidelines), I doubt whether it is reproducible and generalizable.

Thus, while the topic is very relevant and I commend the authors for investigating it, I believe the study can be improved in several aspects. So rather than spending my time commenting on what is already good, I tried to address what could be done better. So I hope my comments may helpful for the authors to accomplish this.

In general

- Please add line numbers to facilitate reviewing.

Abstract

- Title: “Gene Therapy for Alzheimer’s Disease: Clinicians’ perspectives on patient needs” seems to suggest this is about patients need, while only clinicians were interviews. As this does not correspond directly to the findings and is not the best way to study patients’ needs, I suggest changing it.

- Introduction: The aim does not correspond to the aim as specified in the introduction (facilitators are missing)

- Methods: “Personal morals” were not addressed in this study (these generally refer to an individual's principles or standards of behavior) and are therefore not synonymous to religious or spiritual aspects.

- Results: These findings don’t seem to entirely correspond to the headings and findings in the manuscript. And while I can see how patients may find it difficult to understand gene therapy, I would strongly suggest refraining from claiming that they lack the education or ability to do so. These therapies are not introduced in clinical practice yet, the fact that ten clinicians expect that they are unable to comprehend this, does not make it a scientifically supported finding and in my humble opinion this could be perceived as derogatory and offensive, which I am sure was not intended by the authors.

- Discussion/conclusion: I would say clinicians need to be educated themselves and learn to explain this to their patients, rather than merely being aware of their patients’ needs and their own gaps in knowledge.

Introduction

- “that poses social and economic challenges for clinicians, health systems, patients, and families worldwide”: I’d say these are first and foremost challenges for patients, their families, society and the health care system, rather than clinicians.

- The NIA-AA diagnostic framework is intended for research, not clinical practice. and it does not define Alzheimer’s Disease as A+T+N+ but as A+T+. Also, it is not clear to me how biological disease criteria relate to gene therapies.

- How do treatments for sickle cell disease fit in a listing of treatments for AD?

- “E (APOE) gene impart increased genetic risk for AD”: not necessarily, it could also decrease.

- “Every individual carries one of three possible alleles”: they carry two.

- “As rates of AD and AD-related deaths increase in the population, there is high demand from patients, caregivers and patient advocates for effective disease-modifying therapies”: I’d say they ask this because it is a devastating disease and the prevalence is already very high and rising (although according to some reports not as fast as projected), rather the mere fact that deaths are rising.

- “Disparities in dementia care are well documented.” While important, this is a rather abrupt introduction of an entirely new topic.

- “The importance of how clinicians approach complex treatments and clinical decision- making with AD patients and their families cannot be understated.” I suggestion adding a few sentences here and there to smooth these transitions.

- “research exploring how patients and caregivers understand gene therapy as an emerging treatment for AD is lacking.[13] We took a high-level approach, seeking to understand how clinician’s themselves perceive gene therapy and their understanding of patients’ awareness of gene therapy”: asking clinicians is not the best way to study patients perspectives. If you want to study patients’ perspective I would suggest including them in the study population. Or if you only interview clinicians, report clinicians’ perspectives. Again, as far as I know, these gene therapies are not introduced in clinical practice yet, so asking one group what another group may or may not understand about unspecified and unapproved treatments is in my opinion quite an unnecessary stretch.

- I strongly suggest specifying here what is meant by these novel gene therapies, for which patients they are intended, in which stage, with what target, as well as potential benefits and harms, and what exactly may be so difficult to explain and understand. As ApoE is discusses somewhat out of the blue, I gather the authors may have therapies targeting this gene in mind, whereas CRISPR-Cas interventions to alter causal mutations of dominantly inherited Alzheimer’s Disease may introduce a different applications, which do not necessarily include the same views, barriers and facilitators. As this is the topic of this study, it should be specified, at least to some extent. In addition, I would be interested in literature on views, barriers, and facilitators related to gene therapies for other diseases that have already been approved and introduced in clinical practice, as they give insights and a starting point for gene therapies in AD.

Methods:

- “We conducted semi-structured interviews with providers who would play a pivotal role in guiding patients through decisions regarding gene therapy for Alzheimer’s disease.” Please specify this.

- “we collaborated with key health system leaders who suggested clinicians appropriate for the study across disciplines”: again, this is quite vague.

- “The multidisciplinary research team created a semi-structured interview guide complete with open-ended interview questions and probes.”: How? Based on what literature, of which findings? Why include these topics and not others?

- “Three clinicians with social work and neurology expertise in our health system reviewed the guide, and we incorporated their feedback.” What makes them qualified to do so?

- “domains queried in the interview guide included: cost, access, and quality[15]––predominant tensions driving healthcare delivery––as well as patient and clinician concerns around establishing understanding of gene therapy as a treatment. Additionally, we asked about personal morals, including religion and spirituality.”: Why focus on this selection of topics?

- How long were the interviews?

- It would be insightful to include the topic guide and script for the semi-structured interviews.

- How were participants informed about the topic of interest (gene therapies), how was this defined?

Analysis

- “When further observations and analysis were no longer generating new or discrete themes, the study team stopped interviews and recruitment” I find it somewhat unexpected that data saturation was reached after only ten interviews, when they concern such a complex topic.

Results:

- “10 in-depth interviews with clinicians” Please specify what is meant by clinicians. General practitioners or neurologists? In primary care or memory clinics? Academic centers? Did they have experience with gene therapies in clinical trials in this population?

- Sample characteristics: based on this, I don’t think this sample is representative of the physicians who would actually discuss gene therapies with potential patients, given their education and clinical discipline.

- “Analysis resulted in a focus on three main themes with related subthemes – 1) Understanding of gene therapy and Alzheimer’s disease 2) Consideration of inequity (care access, disease awareness along with education level, family support, trust in care systems); and 3) Considerations in decision-making (religious/spiritual beliefs and method of treatment delivery as a decision-making tools).”: How do these correspond to the results reported in the abstract?

Gaps in understanding of gene therapy and Alzheimer’s disease

- “Our clinician interviewees noted that, to provide optimal care to people living with Alzheimer’s disease (AD), healthcare providers should consider the patients’ own preferences.”: Yet everything listed in this paragraph seems to concerned with patients characteristics, rather than the patients’ own preferences?

- Also, how does this relate to understanding of AD and gene therapy?

Consideration of inequity

- “gene therapy––the details of which may even stump a knowledgeable clinician.”: I suggest reconsidering or rephrasing this.

Results in general

These results are very general and rather superficial and seems to apply any new medical therapy for AD, rather than gene therapy specifically.

The findings seem to focus mostly on barriers, where are the facilitators reported…?

In addition, potential patients are referred to as individuals who are cognitively impaired, and potentially unable to perform activities of daily living, to the extent they are no longer competent to make their own decisions. To the best of my knowledge, these therapies are generally intended for patients in earlier stages of the disease, where these challenges play a smaller role and other barriers apply (for example, related to treating relatively healthy individuals with invasive and risky therapies).

Discussion

- “We explored patient needs regarding the future of AD gene therapy from the clinician’s perspective.” Again, I think this study should focus on clinicians’ perspectives alone. Or include patients in the sample population.

- “While colleagues work diligently on the biological science of gene therapy, our team posited that many barriers potentially exist to the successful rollout of gene therapy in the practical world of patient care.”: Our team…?

Conclusion

- “Clinicians working in this area see the challenges ahead and readily suggest what might be done to mitigate them.”: Were these suggestions included in the results?

6. PLOS authors have the option to publish the peer review history of their article (what does this mean?). If published, this will include your full peer review and any attached files.

Reviewer #1: No

Reviewer #2: No

---

## [Author Response · Author response to Decision Letter 0]

14 Jun 2024

Also see formatted attachment with same...

Responses to Reviewers’ Comments to the Author

Thank you for your thoughtful review. We now have a better paper because of this feedback.

Reviewer #1: 

1. Much of your work may be applicable to the use of monoclonal antibodies. Would you consider doing a similar study for that indication? Thank you. Monoclonal antibody treatment to amyloid has proven therapeutic limitations or a therapeutic ceiling of effectiveness. We have described this in the Introduction. Gene therapy actually offers more promise as an effective treatment or possible cure. Therefore, this paper focuses on gene therapy as a potential treatment. 

2. More importantly, your work on gene therapy for AD is more applicable in the near term to APP, PS1 and PS2 mutations. This could be mentioned in your discussion. Consider working with clinicians participating in the DIAN cohort studies and persons at risk of carrying such genes. You may get different answers than from your current study. Thank you for the comment and we feel this is an important point. We added this detail to the introduction and to the discussion sections. 

Reviewer #2: 

Overall

The researchers aimed to understand clinicians’ perspectives regarding gene therapy, their understanding of patients’ views, and potential barriers and facilitators to its use as a future treatment for Alzheimer’s disease (AD).

Given advances in developing these therapies and the lack of knowledge about clinicians’ and patients’ perspective, this is a topical study. It’s important to include the perspectives of the people who might be affected by new innovative treatments at an early stage.

The researchers identified gaps in knowledge, a need for educational materials, and barriers such as cost, difficulty to understand such complex interventions and lack of trust in health care systems.

However, my main concerns are:

1. It remains unspecified what is meant by “gene therapies”, for whom this are intended, in what stage, etcetera. Although I understand that this may vary somewhat, by leaving it so vague, the findings could be applicable to any new medical intervention for Alzheimer’s disease. The introduction was rewritten and expanded to clarify that this paper addressed gene therapies targeted at the APOE4 allele. 

2. The sample population is small (10 health care providers) and not very representative (4 have a Master’s Degree as highest education). Based on this I doubt to what extent these findings may reflect and be generalizable to the perspectives of the clinicians who would discuss gene therapies with patients if and when they are introduced. Thank you for the comment. We addressed the comment in the limitations section of the discussion. 

3. In addition, potential patients are referred to as individuals who are cognitively impaired, and potentially unable to perform activities of daily living, to the extent they are no longer competent to make their own decisions. To the best of my knowledge, these therapies are generally intended for patients in earlier stages of the disease, where these challenges play a smaller role and other barriers apply (for example, related to treating relatively healthy individuals with invasive and risky therapies). Thank you for the comment. We addressed this issue in the limitations section of the Discussion. 

4. Lastly, as this qualitative study is not reported in accordance to the Consolidated criteria for reporting qualitative research (COREQ) checklist or Standards for reporting qualitative research (SRQR) checklist (the authors don’t report adhering to guidelines), I doubt whether it is reproducible and generalizable. We have provided more detail on our adherence to COREQ and have attached the checklist to the resubmission.

Thus, while the topic is very relevant and I commend the authors for investigating it, I believe the study can be improved in several aspects. So rather than spending my time commenting on what is already good, I tried to address what could be done better. So I hope my comments may helpful for the authors to accomplish this. Thank you for your comments. Since the study concluded, we cannot change the methods for this study but we appreciate your comments to incorporate changes in future studies. We did include your point in the limitations section of the Discussion.

In general

- Please add line numbers to facilitate reviewing.

Abstract

5. Title: “Gene Therapy for Alzheimer’s Disease: Clinicians’ perspectives on patient needs” seems to suggest this is about patients need, while only clinicians were interviews. As this does not correspond directly to the findings and is not the best way to study patients’ needs, I suggest changing it. We changed the title of the study to Clinician’s Perspectives on Gene Therapy for Alzheimer’s Disease: A Qualitative Study

6. Introduction: The aim does not correspond to the aim as specified in the introduction (facilitators are missing) Thank you. We corrected this.

7. Methods: “Personal morals” were not addressed in this study (these generally refer to an individual's principles or standards of behavior) and are therefore not synonymous to religious or spiritual aspects. We agree and have revised to be more accurate.

8. Results: These findings don’t seem to entirely correspond to the headings and findings in the manuscript. And while I can see how patients may find it difficult to understand gene therapy, I would strongly suggest refraining from claiming that they lack the education or ability to do so. These therapies are not introduced in clinical practice yet, the fact that ten clinicians expect that they are unable to comprehend this, does not make it a scientifically supported finding and in my humble opinion this could be perceived as derogatory and offensive, which I am sure was not intended by the authors. We have rephrased this as potential gaps in understanding and being sure that clinicians are poised to help inform patients and families, meeting them where they are. We believe we emphasized the need for trust building and rapport in the clinical relationship.

9. Discussion/conclusion: I would say clinicians need to be educated themselves and learn to explain this to their patients, rather than merely being aware of their patients’ needs and their own gaps in knowledge. Thank you for this comment. This comment was added to the first paragraph of the Discussion and we have done our best to emphasize the importance of clinicians’ awareness/education. 

Introduction

10. “that poses social and economic challenges for clinicians, health systems, patients, and families worldwide”: I’d say these are first and foremost challenges for patients, their families, society and the health care system, rather than clinicians. We made this change. 

11. The NIA-AA diagnostic framework is intended for research, not clinical practice. and it does not define Alzheimer’s Disease as A+T+N+ but as A+T+. Also, it is not clear to me how biological disease criteria relate to gene therapies. Thank you for your comments. We state that neurodegeneration criteria is +/-. “To fulfill criteria for pathologic AD, there must be the presence of β-amyloid plaques (A), +/- neurofibrillary tau deposits (T) and +/- neurodegeneration (N).” In clinical practice, pre-clinical diagnosis of AD is made by the presence of amyloid plaques. A clinical diagnosis of AD is made by the presence of B-amyloid plaques in the presence of cognitive decline.

12. How do treatments for sickle cell disease fit in a listing of treatments for AD? The success of gene therapy for the treatment of sickle cell disease is an example of an emerging gene therapy treatment. 

13. “E (APOE) gene impart increased genetic risk for AD”: not necessarily, it could also decrease. APOE4 gives increase in genetic risk; APOE2 decreases genetic risk. We clarified the language in the Introduction to reflect this fact. 

14. “Every individual carries one of three possible alleles”: they carry two. Thank you. We reworded this section to reflect this fact. 

15. “As rates of AD and AD-related deaths increase in the population, there is high demand from patients, caregivers and patient advocates for effective disease-modifying therapies”: I’d say they ask this because it is a devastating disease and the prevalence is already very high and rising (although according to some reports not as fast as projected), rather the mere fact that deaths are rising. We added wording to reflect this idea.

16. “Disparities in dementia care are well documented.” While important, this is a rather abrupt introduction of an entirely new topic. We re-worded this area of the introduction to address this issue.

17. “The importance of how clinicians approach complex treatments and clinical decision- making with AD patients and their families cannot be understated.” I suggestion adding a few sentences here and there to smooth these transitions. We re-worded this section. 

18. “research exploring how patients and caregivers understand gene therapy as an emerging treatment for AD is lacking.[13] We took a high-level approach, seeking to understand how clinician’s themselves perceive gene therapy and their understanding of patients’ awareness of gene therapy”: asking clinicians is not the best way to study patients perspectives. If you want to study patients’ perspective I would suggest including them in the study population. Or if you only interview clinicians, report clinicians’ perspectives. Again, as far as I know, these gene therapies are not introduced in clinical practice yet, so asking one group what another group may or may not understand about unspecified and unapproved treatments is in my opinion quite an unnecessary stretch. We re-worded the introduction to reflect this idea. 

19. I strongly suggest specifying here what is meant by these novel gene therapies, for which patients they are intended, in which stage, with what target, as well as potential benefits and harms, and what exactly may be so difficult to explain and understand. As ApoE is discusses somewhat out of the blue, I gather the authors may have therapies targeting this gene in mind, whereas CRISPR-Cas interventions to alter causal mutations of dominantly inherited Alzheimer’s Disease may introduce a different applications, which do not necessarily include the same views, barriers and facilitators. As this is the topic of this study, it should be specified, at least to some extent. In addition, I would be interested in literature on views, barriers, and facilitators related to gene therapies for other diseases that have already been approved and introduced in clinical practice, as they give insights and a starting point for gene therapies in AD. Thank you for this comment. We addressed this issue in the introduction and the discussion. 

Methods: Please address these comments for methods, analysis, results

20. “We conducted semi-structured interviews with providers who would play a pivotal role in guiding patients through decisions regarding gene therapy for Alzheimer’s disease.” Please specify this. We revised to “We conducted semi-structured interviews with academic center clinicians who would guide patients through decisions regarding gene therapy for Alzheimer’s disease.”

21. “we collaborated with key health system leaders who suggested clinicians appropriate for the study across disciplines”: again, this is quite vague. We changed this to “To recruit clinicians, we collaborated with department heads who suggested clinicians appropriate for the study across disciplines, those who would usually work with patients in the setting of Alzheimer’s disease.”

22. “The multidisciplinary research team created a semi-structured interview guide complete with open-ended interview questions and probes.”: How? We have the additional clarifying text “Three clinicians with social work and neurology expertise, knowledge of Alzheimer’s disease, and knowledge of research methods in our health system reviewed the guide and helped the team identify domains of interest to our health system at the time. NAB piloted the interview guide with two clinicians to improve understanding and word clarity. We incorporated feedback from both engagements. Along with demographic questions, domains queried in the interview guide included: cost, access, and quality[15]––predominant tensions driving healthcare delivery––as well as potential patient and clinician concerns around establishing understanding of gene therapy as a treatment. Additionally, we asked about the role of religion and spirituality, as our interview guide reviewers indicated these as important considerations in patients’ health decision making locally.”

23. “Three clinicians with social work and neurology expertise in our health system reviewed the guide, and we incorporated their feedback.” What makes them qualified to do so? The additional qualification was added to the paragraph and is reflected in the excerpt above under response 22.

24. “domains queried in the interview guide included: cost, access, and quality[15]––predominant tensions driving healthcare delivery––as well as patient and clinician concerns around establishing understanding of gene therapy as a treatment. Additionally, we asked about personal morals, including religion and spirituality.”: Why focus on this selection of topics? We included topics that might impact a patient or clinician’s decision to pursue gene therapy as a treatment as identified by knowledgeable clinicians at our academic center. While the effectiveness of a treatment’s symptom-mitigating or disease-altering quality is of paramount concern, a patient could choose not to move forward with that treatment if the clinic where it is provided is too far away or if the treatment is too expensive. When a patient understands the treatment they are considering––including the risks and benefits of undergoing that treatment––they may be more inclined to choose it. Additionally, more personal factors like individual belief systems, religion, and spirituality may impact why a patient chooses one treatment over another – as indicated by our collaborators. Our study aims to better understand how these factors might interact in the context of using gene therapy to treat Alzheimer’s disease. 

25. How long were the interviews? The interview guide comprised 10 questions, and each interview lasted between twenty minutes to an hour. Variability in the length of interview depended on the individual care provider’s communication style and follow questions. This detail on time was added to the first line of Results.

26. It would be insightful to include the topic guide and script for the semi-structured interviews. Thank you for your comment. We will add the interview guide as a supplementary document.

27. How were participants informed about the topic of interest (gene therapies), how was this defined? Great point. Participants were first asked about their current level of understanding of gene therapy. Interviewers then followed this initial question by defining gene therapy for each participant with the same standard definition: “Gene therapy is an experimental technique that uses genes - a unit of heredity written in our DNA - to treat or prevent disease. In the future, this technique may allow health care providers to treat a disorder, such as Alzheimer’s disease by adding or changing a gene in a patient's cells.” This detail was added to Measurement.

Analysis

28. “When further observations and analysis were no longer generating new or discrete themes, the study team stopped interviews and recruitment” I find it somewhat unexpected that data saturation was reached after only ten interviews, when they concern such a complex topic.

Results:

29. “10 in-depth interviews with clinicians” Please specify what is meant by clinicians. General practitioners or neurologists? In primary care or memory clinics? Academic centers? Did they have experience with gene therapies in clinical trials in this population? These clinicians were academic center clinicians working with patients in the setting of Alzheimer’s disease diagnosis or ongoing care and have clarified this in the Results section.

30.

---

## [Decision Letter · Decision Letter 1]

24 Jun 2024

PONE-D-24-09705R1Clinician’s Perspectives on Gene Therapy for Alzheimer’s Disease: A Qualitative StudyPLOS ONE

Dear Dr. Boucher,

Thank you for resubmitting your work to PLOS ONE. Please make the corrections posed by Reviewer #2 so I can render a decision on this manuscript.

**Comments to the Author**

1. If the authors have adequately addressed your comments raised in a previous round of review and you feel that this manuscript is now acceptable for publication, you may indicate that here to bypass the “Comments to the Author” section, enter your conflict of interest statement in the “Confidential to Editor” section, and submit your "Accept" recommendation.

Reviewer #1: All comments have been addressed

Reviewer #2: (No Response)

2. Is the manuscript technically sound, and do the data support the conclusions?

Reviewer #1: Yes

Reviewer #2: Partly

3. Has the statistical analysis been performed appropriately and rigorously? 

Reviewer #1: N/A

Reviewer #2: Yes

4. Have the authors made all data underlying the findings in their manuscript fully available?

Reviewer #1: Yes

Reviewer #2: No

5. Is the manuscript presented in an intelligible fashion and written in standard English?

Reviewer #1: Yes

Reviewer #2: Yes

6. Review Comments to the Author

Reviewer #1: Thank you answering my questions. Hopefully you will follow-up on this topic in the future. Consider reaching out to the DIAN network.

Reviewer #2: Dear authors,

Although most of my comments have been addressed appropriately, I still have one major issue:

The introduction has been rewritten and expanded to clarify that this paper addresses clinician's views on gene therapies targeted at the APOE4 allele.

However the methods section now states that participants were provided the following - very general - definition of gene therapies: "Gene therapy is an experimental technique that uses genes - a unit of heredity written in our DNA - to treat or prevent disease. In the future, this technique may allow health care providers to treat a disorder, such as Alzheimer’s disease by adding or changing a gene in a patient's cells."

Yet ApoE4 is not mentioned anywhere in the interview guide. Moreover, most questions, except for the first one, relate to "new Alzheimer’s care and treatment strategies", or "a treatment is able to stop Alzheimer’s" or "an Alzheimer’s treatment". Some do not even mention treatment of any kind ("6. According to the Alzheimer’s Association, Black Americans are diagnosed with Alzheimer’s at a later stage than White Americans. a. Why do you think this happens?"). The interview guide does not seem to focus on gene therapies targeted at ApoE4, or even gene therapies in general, but at any kind of hypothetical new strategy for treatment or care.

I think this is problematic, as the clinicians who were interviewed, seemed to talk about treatments in a very broad sense, given their comments about patients' decision-making capacities, assuming they are cognitively impaired, even confined to bed, or have “progressed to such a degree that it seems that any further treatment would just be burdensome”. In my opinion it is extremely unlikely that gene therapies would be offered to patients with MCI or dementia. So I get the impression that the results do not relate to gene therapies targeted at ApoE4, but to any kind of treatment or care in any stage of the disease.

7. PLOS authors have the option to publish the peer review history of their article (what does this mean?). If published, this will include your full peer review and any attached files.

**Do you want your identity to be public for this peer review?** For information about this choice, including consent withdrawal, please see our Privacy Policy.

Reviewer #1: No

Reviewer #2: No

We look forward to receiving your revised manuscript.

Kind regards,

Stephen D. Ginsberg, Ph.D.

Section Editor

PLOS ONE
---

## [Author Response · Author response to Decision Letter 1]

1 Jul 2024

Reviewer #2: Dear authors,

Although most of my comments have been addressed appropriately, I still have one major issue:

The introduction has been rewritten and expanded to clarify that this paper addresses clinician's views on gene therapies targeted at the APOE4 allele.

However the methods section now states that participants were provided the following - very general - definition of gene therapies: "Gene therapy is an experimental technique that uses genes - a unit of heredity written in our DNA - to treat or prevent disease. In the future, this technique may allow health care providers to treat a disorder, such as Alzheimer’s disease by adding or changing a gene in a patient's cells."

Yet ApoE4 is not mentioned anywhere in the interview guide. Moreover, most questions, except for the first one, relate to "new Alzheimer’s care and treatment strategies", or "a treatment is able to stop Alzheimer’s" or "an Alzheimer’s treatment". Some do not even mention treatment of any kind ("6. According to the Alzheimer’s Association, Black Americans are diagnosed with Alzheimer’s at a later stage than White Americans. a. Why do you think this happens?"). The interview guide does not seem to focus on gene therapies targeted at ApoE4, or even gene therapies in general, but at any kind of hypothetical new strategy for treatment or care.

I think this is problematic, as the clinicians who were interviewed, seemed to talk about treatments in a very broad sense, given their comments about patients' decision-making capacities, assuming they are cognitively impaired, even confined to bed, or have “progressed to such a degree that it seems that any further treatment would just be burdensome”. In my opinion it is extremely unlikely that gene therapies would be offered to patients with MCI or dementia. So I get the impression that the results do not relate to gene therapies targeted at ApoE4, but to any kind of treatment or care in any stage of the disease.

RESPONSE: Thank you for these additional comments. We changed the final paragraph of the introduction and the first paragraph of the discussion to address the issue of addressing views specifically on gene therapy with APOE4 which as you point out is in error. We agreed with your comment. We changed the language to mention the possible gene therapies with AD, but to state that the study addressed general gene therapy for AD. We agree that we assess views on novel treatments, but this is in the context of gene therapy as a novel treatment. We feel it is important to gauge clinician decision making of novel treatments in the MCI/dementia due to AD population. Many novel therapies for AD, including gene therapy will most likely be tested or trialed in a cognitively impaired population before testing in a cognitively normal population, for safety reasons. We state this in the conclusion. We thank you for your review and opportunity to improve our paper on these important ideas related to possible future AD treatments.

---

## [Editor Report · Decision Letter 2]

9 Jul 2024

Clinician’s Perspectives on Gene Therapy for Alzheimer’s Disease: A Qualitative Study

PONE-D-24-09705R2

Dear Dr. Boucher,

We’re pleased to inform you that your manuscript has been judged scientifically suitable for publication and will be formally accepted for publication once it meets all outstanding technical requirements.

Kind regards,

Stephen D. Ginsberg, Ph.D.

Section Editor

PLOS ONE

---

## [Editor Report · Acceptance letter]

10 Jul 2024

PONE-D-24-09705R2 

PLOS ONE

Dear Dr. Boucher, 

I'm pleased to inform you that your manuscript has been deemed suitable for publication in PLOS ONE. Congratulations! Your manuscript is now being handed over to our production team.

Kind regards, 

on behalf of

Dr. Stephen D. Ginsberg 

Section Editor

PLOS ONE